# Nonlinear Impact of Circulation-Industry Intelligentization on the Urban–Rural Income Gap: Evidence from China

**Haoyun Meng** [1,*] **, Peidong Deng** [1] **and Jinbo Zhang** [2]

1 School of Economics and Finance, Xi'an Jiaotong University, Xi'an 710061, China; 18966708326@163.com
2 College of Environmental Science and Engineering, Peking University, Beijing 100871, China; kimballzhang@outlook.com
* Correspondence: yun1996@stu.xjtu.edu.cn

**Abstract:** Integrating informatization into the circulation industry has led to the concept of circulation-industry intelligence. By reducing transportation costs and increasing total factor productivity, the incomes of rural-area residents can be improved; a new pattern of regional economy can be established; urban, rural, social, and economic development can become more coordinated; and social sustainable development can be promoted. In this study, we used China's provincial panel data corresponding to the 2007–2019 period to measure the intelligence index of the circulation industry in each region and determine the factors that affect the urban–rural income gap; thereafter, we conducted comparative analyses. Further, a fixed-effects model was established based on the theory of agglomeration and diffusion effects to analyze the relationship between these two variables. Our analysis identified innovation investment as a significant intermediary mechanism. The robustness of this finding was verified by substituting variables and controlling for endogeneity. Thus, the effect was shown to be regionally heterogeneous. This study innovatively integrated informatization into the circulation industry, and the results obtained provide a reference for formulating transportation infrastructure as well as informatization strategies for promoting urban–rural coordination and sustainable development globally.

**Keywords:** circulation-industry intelligence; urban–rural income gap; heterogeneity analysis; innovation investment; intermediary effect

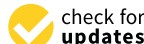



## 1. Introduction

The sustainable development of China's economy is facing several challenges due to the COVID-19 pandemic and international wars. These exogenous factors exert different effects on the income levels of residents in different regions owing to differences in transportation and information infrastructure between urban and rural areas. Narrowing the income gap between these two areas, which represents an important strategy for promoting sustainable economic development, can promote the coordinated development of these areas [1]. In 2002, the per capita income of urban residents in China was more than three-fold higher that of rural residents; however, in 2021 this ratio decreased to two-and-a-half-fold [2]. Further, the development of the circulation industry can narrow the urban–rural income gap through channels such as improved transportation efficiency and costs, and can effectively promote sustainable economic development; however, the output value of the traditional circulation industry is gradually decreasing. In 2002, the added value of transportation in China accounted for 6.16% of the national economy, but in 2021 this added value decreased to 4.11% (data are calculated from data provided by the National Bureau of Statistics of China).

Although the development of the circulation industry has accelerated economic flow, it has also had a negative impact on the environment owing to the associated carbon emissions [3–5] and water pollution [6,7]. Additionally, attaining environmental sustainability

and economic development often seems contradictory [8]. It has also been reported that in this information age, a combination of the internet and the real economy plays a greater role in advancing the economy [9,10]. Further, the development of information technology has also alleviated the impact of the circulation industry on carbon emissions [11,12], and the integration of the information industry into the circulation industry constitutes the concept of circulation-industry intelligentization. Furthermore, the development of circulation-industry intelligence can provide new ideas for enhancing and promoting sustainable economic development and can also contribute strategies for developing the economies of resource-poor areas. By narrowing the income gap between urban and rural areas, the development of circulation-industry intelligence can also lead to a balance in the speed of urban and rural development as well as coordinated sustainable social development in these areas. However, the current research is limited to the traditional circulation industry, and there are no studies on the combined impact of the circulation industry and informatization on the urban–rural income gap. Therefore, it is necessary to investigate how this urban–rural income gap can be narrowed via the establishment of an intelligent circulation industry. The results of such a study would also be of great significance in promoting sustainable development in both regions. Additionally, as the world's second largest economy, these results for China can provide a wide range of references that can be applied in other parts of the world.

China has one of the highest urban–rural income gaps in the world [13], and there are several environmental [14], economic, and macro factors that affect this income gap, such as air quality [15,16], water resources [6,7], housing prices [17,18], city size [19], labor mobility [20], e-commerce [21], and digital financial inclusion [22]. As the basic industry for regional development, the circulation industry has a significant impact on the urban–rural income gap; however, whether the benefits brought about by the development of this industry to the economy can be equally shared by urban and rural areas remains unclear [23].

Presently, there is no clear definition regarding circulation-industry intelligentization; however, most scholars agree that the integration of information into transportation in this digital economy era can impact the economy [23–25]. Additionally, the development of circulation-industry intelligence can also play an important role in the establishment of green urban economic systems and promote sustainable development [26]. Notably, the development of the traditional circulation and transportation industries also affects economic development via the improvement of infrastructure, and in this information age, digital information systems can provide more possibilities for the development of the transportation and circulation industries [27–29]. Thus, it has been observed that the intelligentization of the circulation industry can reduce the urban–rural income gap by improving income distribution, while promoting the economy. Further, the integration of information into the circulation industry can lead to a decrease in the negative impact on urban traffic (in terms of congestion) and improve the efficiency of urban travel [30]. It has also been reported that the intelligentization of the circulation industry can promote the establishment of economies of scale in cities as well as regional sustainable development [31].

The infrastructure of the circulation and transportation industries is the advanced capital for social development, and in particular, the circulation industry has an important impact on income distribution. The development of circulation-industry development also offers the possibility to increase the income and sales volume of products in rural areas, thus promoting the development of e-commerce [31]. The improvement of infrastructure can also promote the flow of labor by providing more employment opportunities as well as productivity in rural areas, thereby balancing urban and rural income levels [32,33]. However, there are significant differences between urban and rural areas in China with respect to the informatization of circulation infrastructure. Further, compared with rural areas, urban areas have geographical and salary advantages that attract IT talents [34,35]. Thus, the impact of the development of circulation-industry intelligence on the income gap between urban and rural areas in China is still uncertain. Most previous studies have

been focused on the impact of traditional circulation or infrastructure differences in the transportation industry on income inequality [36–38]. For example, in a previous study, the impact of constructing high-speed railways on the regional economy and income distribution was investigated [39,40]. However, research on the incorporation of informatization within the circulation industry and the effects on income gaps is limited.

In addition to its effect on regional economic development, differences in the levels of circulation-industry infrastructure development between urban and rural areas also bring about differences in foreign investment. The primary reason for this is that the decline in transportation costs reduces production costs and encourages companies to invest locally [41]. It has also been reported that in some developing countries, improving railways and roads plays a more significant role in attracting foreign investment than improving the aviation industry [42]. Further, the impact of the investment of natural capital on energy efficiency is non-linear, i.e., the same level of investment in rural and urban areas results in income differences between the two regions [43]. However, these differences can be bridged considering that the arrival of the information age can provide a foundation for the integration of information into the traditional circulation industry; moreover, the popularization and application of the internet can also improve the ability of different regions to attract investment [44]. In economically developed regions, the effects of positive externalities created by the internet can facilitate the attraction of foreign direct investment (FDI). Conversely, in economically underdeveloped regions, the effects of negative externalities created by the internet can hinder FDI [45]. Due to differences in the rate of economic development between urban and rural areas, there are gaps in the overall development of the circulation industry. However, informatization that can attract different investments; this could be the reason for the differences in income levels between urban and rural residents.

In this study, we primarily focused on the impact of the development of circulation-industry intelligence on the income gap between urban and rural areas in China, and also examined whether innovation investment could be an effective intermediary mechanism in narrowing this gap. In other words, we investigated whether the informatization of the circulation industry via innovation investments can eventually lead to a decrease in the urban–rural income gap. Taking existing literature into consideration, the contribution of this study is mainly reflected by the following three points: (1) the present study is not limited to the traditional circulation industry, but innovatively explores the impact of the development of the informatization of China's circulation industry on the urban–rural income gap; (2) at the theoretical level, this study provides an analysis of the impact of China's circulation-industry intelligence on the urban–rural income gap in terms of the agglomeration and diffusion effects, and reasonably explains the heterogeneity of the impact at different stages; and (3) this study also highlights innovation investment as a mediating factor in the model to further test the impact of circulation-industry intelligence on the urban–rural income gap. Therefore, the results obtained can provide reliable suggestions for the development of the circulation industry in this age of information and digital economy. Further, to ensure accuracy, we conducted robustness checks on all the results obtained.

## 2. Study Design

### 2.1. Research Ideas

The intelligent development of the circulation industry will have an opposite impact on innovation investment and the urban–rural income gap due to the agglomeration effect and diffusion effect. This section presents the theoretical impact of this study.

### 2.1.1. Agglomeration Effect

The agglomeration effect, initially proposed by Porter [46], believes that when a large number of closely related economic activities gather in a certain area, a cluster advantage will be generated continuously for a certain period of time. Since the improvement in circu-

lation infrastructure has improved the efficiency of the flow of production factors in regions, it is clear that the development of the transportation and circulation industries has promoted the formation of industrial agglomeration [47]. In addition, industrial agglomeration can improve the location advantage of regional economic development, thereby affecting income. Therefore, differences in the levels of industrial agglomeration between regions will cause income gaps. These differences are affected by firm size [48], traffic density [49], economic policy [50], industrial protection policy [51], regional development [52,53], etc.

Generally, the agglomeration effect caused by the circulation industry will increase the income gap between urban and rural areas [54]. Differences in trade cost and location advantages [55] between urban and rural areas promote the flow of production factors, thereby increasing the urban–rural income gap. Picard & Zeng [56] illustrated this from an agricultural perspective. Their research argues that when the transportation cost of agricultural products is high, the income of residents that mainly work in agriculture will decrease, which accelerates the flow of labor from rural to urban areas. It is worth noting that the impact on this flow is more pronounced in more economically developed regions [57]. Zhao et al. [58] took producer services as their research object. They found that the agglomeration effect of transportation, and warehouse and postal services had the greatest impact when studying the impact of industrial agglomeration on the urban–rural income gap. This conclusion further proves the agglomeration effect of the circulation industry.

Agglomeration effects also have a significant impact on investment. Industrial agglomeration increases the degree of technical specialization [59], whereby a higher degree of specialization can attract more investment and enterprises [60]. When the regional circulation industry begins to develop informatization at a lower basic level, labor and other means of production will gather in regions with easier access to information and higher investment returns [61], thus exacerbating the income gap. The impact of investment on industrial agglomeration is affected by market structure, firm size [62], and the degree of foreign trade [63]. In particular, foreign direct investment increases the spatial externality of cities, and the industrial development of cities will improve the regional industrial layout and structure due to knowledge spillovers brought by investment [64]. Investment is an important measure to balance regional incomes, but the comparative advantage generated by the biased investment policies of underdeveloped regions is unsustainable, and may also result in a spatial mismatch of resources [65]. To summarize, investment may be an intermediary mechanism through which the agglomeration effect acts on the circulation industry and affects the urban–rural income gap.

### 2.1.2. Diffusion Effect

The development of the circulation industry will improve the economic development level and residents' incomes in urban areas due to the agglomeration effect, but this promotion is not unlimited. Since the marginal effect is diminishing, an upper limit of the economic dividend is brought by the agglomeration effect. When the urban–rural income gap reaches a critical value, continuing to improve the facilitation level of the circulation industry will result in diseconomies of scale in urban areas. More production flow causes rural areas to have higher circulation costs and causes urban areas to have lower circulation costs; thus, negative results such as rising land rents in industrial-agglomeration areas and a slowing-down of the rate of human capital accumulation will occur. It will also aggravate the traffic congestion and environmental pollution in urban areas [8], which is not conducive to the sustainable development of the urban economy. At the same time, the development of the circulation industry will have a diffusion effect. With the influence of the diffusion effect, the urban economy will spread to the surrounding rural areas, accelerating the economic development of the surrounding areas, thereby reducing the income gap [66].

It is generally believed that the diffusion effect decreases with increasing distance between regions [67,68]. In addition, the promotion efficiency of different industries caused by the diffusion effect of technology will be affected by industry policies [69]. The diffusion

of technology from technologically advanced areas (usually urban) to less technologically advanced areas (usually rural) can alleviate income inequality by attracting labor and investment. Compared with the investment channel, the labor channel brings more benefits to workers and has a more significant effect on reducing the income gap [70]. In addition to the diffusion effect caused by the circulation industry, the development of Internet technology has also had a significant impact on income [71,72]. The development of information technology has promoted an improvement in productivity, thereby increasing the income levels of residents. In addition to raising income levels, the popularity of the Internet also helps regions attract foreign direct investment, thereby reducing the urban–rural income gap [73,74]. The development of the Internet and information technology can reduce the cost of information on production areas and sales areas by improving the connection between rural areas and urban areas, and provide a technical foundation for narrowing the urban–rural income gap. In addition, the proliferation of e-commerce due to the Internet also has a positive impact on national development [75]. Therefore, the diffusion effect also has a significant impact on the influence of the informatization of the circulation industry on investment and the urban–rural income gap.

Overall, the research ideas of this paper are shown in Figure 1.

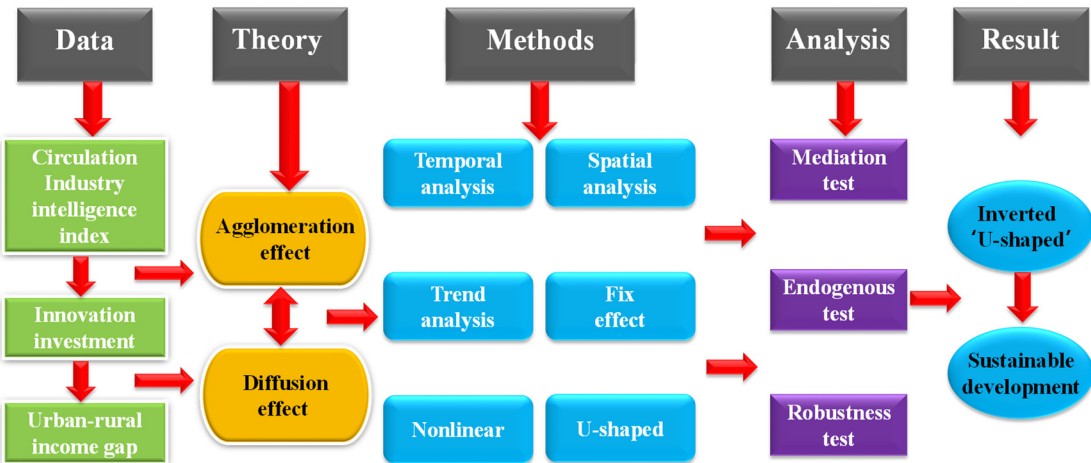

**Figure 1.** The research ideas.

### 2.2. Variable Description

2.2.1. Dependent Variable

Before analyzing the impact of the intelligent development of the circulation industry on the urban–rural income gap, this study, firstly, set the rural per capita disposable income (countryincomeit) and urban per capita disposable income (cityincomeit) as the explained variables for regression. By analyzing the regression results, we observed the difference in the impact of the intelligentization of the circulation industry on per capita disposable income in urban and rural areas. Then, a hypothesis stating that the intelligent development of the circulation industry affects the urban–rural income gap was obtained. This research used the Theil index as a surrogate indicator of the explained variable. Its calculation method is shown in Formula (1).

$$theil_{it} = \sum_{j=0}^{1} \frac{PI_{j,it}}{PI_{it}} \times \ln \left[ \frac{\frac{PI_{j,it}}{PI_{it}}}{\frac{L_{j,it}}{L_{it}}} \right] \tag{1}$$

In Formula (1), $j = 0$ represents rural areas and $j = 1$ represents urban areas. $PI_{it}$ represents the sum of the total annual disposable income of the average per capita households in urban areas and rural areas in year $t$ in region $i$. $PI_{j,it}$ represents the average annual

disposable income of households in rural areas and urban areas, respectively. Similarly, $L_{it}$ represents the sum of the total population of urban and rural households in area $i$ in year $t$. $L_{j,it}$ represents the number of households in rural areas and urban areas, respectively. To summarize, the Theil index of region $i$ in year $t$, that is, the urban–rural income gap, is $theil_{it}$.

### 2.2.2. Independent Variable

The main explanatory variable of this study is the intelligence level of the circulation industry. The intelligentization of the circulation industry needs to consider many factors. this study draws on the construction method of Sun et al. [76]. To summarize, the system of explanatory variables constructed in this study is shown in Table 1.

**Table 1.** Construction of the intelligent system of the circulation industry.

| Categories | Indicators |
| --- | --- |
| Infrastructure | Intelligent human resource investment. |
| | Investment in intelligent equipment. |
| | Software popularization and application. |
| Service | Information resource collection capability. |
| | Data-processing and storage capabilities of digital platforms. |
| | Operation and maintenance of relevant network platforms. |
| Economic | Labor productivity in logistics industry. |
| Environment | Energy consumption. |

Infrastructure investment in networks and transportation forms the basis of the intelligentization of the circulation industry. The infrastructure related to the intelligence of the circulation industry is mainly measured from two perspectives: labor and capital. The labor force is mainly represented by the input of intelligent human resources. Since there are no provincial-level data for the educational composition of China's circulation practitioners, this study used the proportion of the number of people with higher education in each province multiplied by circulation practitioners as a proxy variable. The proportion of people with higher education was calculated using the number of undergraduate students and undergraduate graduates divided by the local employment population for the year. Capital was represented by the investment in intelligent equipment. The measurement index selected for the variable was the proportion of the import value of the electronic-information manufacturing industry to the added value of the circulation industry. The import value of the electronic-information manufacturing industry included both non-tax-refundable goods (import and export mark 1) and duty-free goods (import and export mark 2). Owing to the incomplete statistical data on the added value of the circulation industry by region, only some county-level cities had data; therefore, the added value of the transportation, warehousing, and postal industries was used instead.

The evaluation of the intelligent-service level of the circulation industry was the crux of this study. The indicators included in the variables needed to fully consider the results of the integration of artificial-intelligence technology with the circulation industry, and also evaluate the intelligent-service level of the circulation industry from multiple perspectives such as market demand, information decision-making, and risk assessment. At this level, the popularization and application of software in China should be considered first; therefore, the measurement index that was adopted to achieve this was the proportion of software product revenue to the added value of the circulation industry. The second consideration should be the information-resource collection ability of the circulation industry; therefore, to analyze this, the mobile phone penetration rate in each province was used as a proxy indicator. The third consideration should be the data-processing and storage capabilities of the digital platform of the circulation industry; therefore, an indicator measuring the proportion of data-processing and storage-service revenue in the added value of the circulation industry was used (which included revenue from information-technology

services, information-technology value-added services, information-technology consulting services, information-system integration, and support services). Finally, the operation and maintenance of the network platform related to the circulation industry should also be considered; this indicator measured the proportion of each province's data-processing and operation-service revenue and information-security revenue to the added value of the transportation industry.

In terms of the economic benefits of the intelligent development of the circulation industry, the measurement index used in this study was the labor productivity of China's circulation industry. Total labor productivity was calculated by dividing the industrial added value of industrial enterprises by the average number of all employees in the same period. This indicator also considers the relationship between labor and economic benefits.

Based on the requirements of sustainable development, environmental benefits were also prioritized in the process of developing the intelligentization of the circulation industry. The environmental benefits were measured via the energy consumption of the circulation industry in each province. This indicator was first calculated using the ratio of the total energy consumption of the circulation industry to the total energy in that year, and then, assigned to each province.

In this research, the measurement of the intelligence level of the circulation industry adopts the global entropy method, which considers both time and space as influencing factors. The explanatory variables of the region $i$ in the year $t$ obtained by the entropy method are finally expressed as $incir_{it}$.

### 2.2.3. Mediating Variable

The mediating variable selected in this study was the proportion of innovation investment measured as the proportion of the RD expenditure of industrial enterprises above a designated size in each province to the regional GDP. The proportion of innovation investment in region $i$ in year $t$ is denoted as $ri_{it}$.

### 2.2.4. Control Variables

The control variables were calculated as follows: First, it is important to note that the speed of economic development (*lgdp*) affects the urban–rural income gap. To mitigate the effect of the time trend, the logarithm of the GDP index of each province in China was used to represent this indicator. Second, the development speed of e-commerce (*pk*) affects the sales volume of goods in various regions, which, in turn, affects the income of residents. Therefore, this study used the number of express deliveries per capita to represent the development speed of e-commerce to control the impact of the intelligentization of the circulation industry on the urban–rural income gap. Third, the foreign trade indicator (*lexgdp*) represented the development of the local economy and industry, which has an impact on the urban–rural income gap; therefore, to reduce the influence of the time trend, the logarithm of the ratio of the total import and export of the business unit to the regional GDP was used to represent this indicator. Fourth, the infrastructure-investment variable (*lingdp*) also needed to be controlled, and the logarithm of the ratio of the completion of transportation fixed-asset investment to the regional GDP was therefore used to represent this indicator. Finally, the development of information technology (*linnogdp*) affects regional economic development and income distribution, and the logarithm of the ratio of local fiscal science and technology expenditure to regional GDP was used to represent this variable.

This research analyzes the panel data of 31 provincial-level administrative regions in China from 2007 to 2019 to test the impact of the intelligent development of the circulation industry on the income gap between urban and rural areas in China. All data involved in this article come from the *China Statistical Yearbook*, the *China Tertiary Industry Statistical Yearbook*, the *China Electronic Information Industry Statistical Yearbook*, the *China Information Industry Yearbook*, the *China Environmental Statistical Yearbook*, the *China Energy Statistical Yearbook*, the CSMAR Database, the Guoyan Network, and the EPS Global Statistics data-analysis platform.

In summary, all the variables in this study are shown in Table 2.

**Table 2.** Variable Description.

| | **Dependent Variable** | |
|---|---|---|
| Urban–rural income gap | Theil index | $theil_{it}$ |
| | Independent variable | |
| The intelligence level of the circulation industry | Weighted score calculated via entropy method | $incir_{it}$ |
| | Mediating variable | |
| Proportion of innovation investment | R&D expenditure of industrial enterprises above designated size as a percentage of regional GDP | $ri_{it}$ |
| | Control variables | |
| Economic development | The logarithm of the GDP index for each province | $lgdp_{it}$ |
| The development of e-commerce | Number of express deliveries per capita | $pk_{it}$ |
| Foreign trade | The logarithm of the ratio of the total import and export of the business unit to the regional GDP | $lexgdp_{it}$ |
| Infrastructure investment | The logarithm of the ratio of the completion of transportation fixed-asset investment to the regional GDP in each region | $lingdp_{it}$ |
| Information-Technology Development | The logarithm of the ratio of local financial science and technology expenditure to regional GDP | $linnogdp_{it}$ |

### 2.3. Model Design

This research first analyzes the impact of the intelligent circulation industry on the income of residents in China's urban areas and the impact on the intelligent development of the circulation industry on the income of residents in China's rural areas. The regression equations are shown in Equations (2) and (3).

$$cityincome_{it} = \alpha_0 + \alpha_1 incir_{it} + \sum_{j=1}^{5} \alpha_j X_{j,it} + \mu_i + \varepsilon_{it} \tag{2}$$

$$countryincome_{it} = \alpha_0 + \alpha_1 incir_{it} + \sum_{j=1}^{5} \alpha_j X_{j,it} + \mu_i + \varepsilon_{it} \tag{3}$$

Before the level of intelligence in the circulation industry reaches the threshold, the higher the level of intelligence on the circulation industry, the higher the income gap between urban and rural areas. However, after the intelligence level of the circulation industry reaches the threshold, with improvement in the intelligence level of the circulation industry, the income gap between urban and rural areas should be gradually reduced. On this basis, the following assumptions are made:

**Hypothesis 1.** *There is an inverted "U"-shaped relationship between the intelligence of the circulation industry and the urban–rural income gap.*

This research adds the quadratic term of the intelligentization index of the circulation industry to measure the difference in the influence of the intelligentization of the circulation industry at different stages. In order to distinguish the heterogeneous impact of the intelligentization of the circulation industry on the urban–rural income gap in 31 regions of China, this paper adopts the fixed effect as the main model, and the benchmark regression equation is shown in Equation (4).

$$theil_{it} = \alpha_0 + \alpha_1 incir_{it}^2 + \alpha_2 incir_{it} + \sum_{j=1}^{5} \alpha_j X_{j,it} + \mu_i + \varepsilon_{it} \tag{4}$$

The income of residents in urban areas of region $i$ in year $t$ is denoted as *cityincome*$_{it}$, the income of residents in rural areas is denoted as *countryincome*$_{it}$, and the urban–rural income gap is denoted as *theil*$_{it}$. The core explanatory variable of this paper is the degree of intelligent development of China's circulation industry. The degree of intelligent development of the circulation industry in region $i$ in year t is expressed in the text by *incir*$_{it}$. There are five control variables represented by $X_{j,it}$. $\alpha_j$ is the coefficient representing the $j$th control variable for region $i$ in year $t$. $\mu_i$ represents the fixed effect of region, and $\varepsilon_{it}$ is the random-error term of region $i$ in year $t$.

In part of the intermediary mechanism test, the index variable of innovation investment is introduced. The proportion of innovation investment in region $i$ in year $t$ is denoted by $ri_{it}$. The mediation mechanism test is divided into three steps: the first step coincides with the benchmark regression to verify the inverted "U"-shaped relationship between the explanatory variable *incir*$_{it}$ and the explained variable *theil*$_{it}$. Second, in the theoretical mechanism part, it is assumed that the influence of the proportion of innovation investment on explanatory variables is also an inverted "U" shape, so it is necessary to introduce the quadratic term of the degree of intelligence of the circulation industry for judgment. Accordingly, Hypothesis 2 is obtained:

**Hypothesis 2.** *There is an inverted "U"-shaped relationship between the intelligence of the circulation industry and the proportion of innovation investment.*

The fixed-effects model established in this step is shown in Equation (5).

$$ri_{it} = \alpha_0 + \alpha_1 incir_{it}^2 + \alpha_2 incir_{it} + \sum_{j=1}^{5} \alpha_j X_{j,it} + \mu_i + \varepsilon_{it} \tag{5}$$

The third step is to verify the positive correlation between the mediating effect variable $ri_{it}$ and the explained variable *theil*$_{it}$. In the process of mediating mechanism testing, in addition to verifying that the effects of the explanatory variables on the mediating variables and the explanatory variables on the outcome variables are the same, it is also necessary to ensure that the correlation between the mediating variables and the explained variables is positive. In other words, regional innovation investment and the urban–rural income gap change to follow the same direction. This leads to Hypothesis 3:

**Hypothesis 3.** *The proportion of innovation investment changes to follow the same direction as the urban–rural income gap.*

This step still follows the fixed-effects model, and the specific representation is shown in Equation (6).

$$theil_{it} = \alpha_0 + \alpha_1 ri_{it} + \sum_{j=1}^{5} \alpha_j X_{j,it} + \mu_i + \varepsilon_{it} \tag{6}$$

## 3. Spatiotemporal Analysis and Empirical Analysis

### 3.1. Spatial Analysis of the Urban–Rural Income Gap in China

First, we examined the trend changes in the income gap between urban and rural areas in China from 2007 to 2019. Figure 2 shows the average change in the urban–rural income gap (represented by m_theil) in each region in China every year.

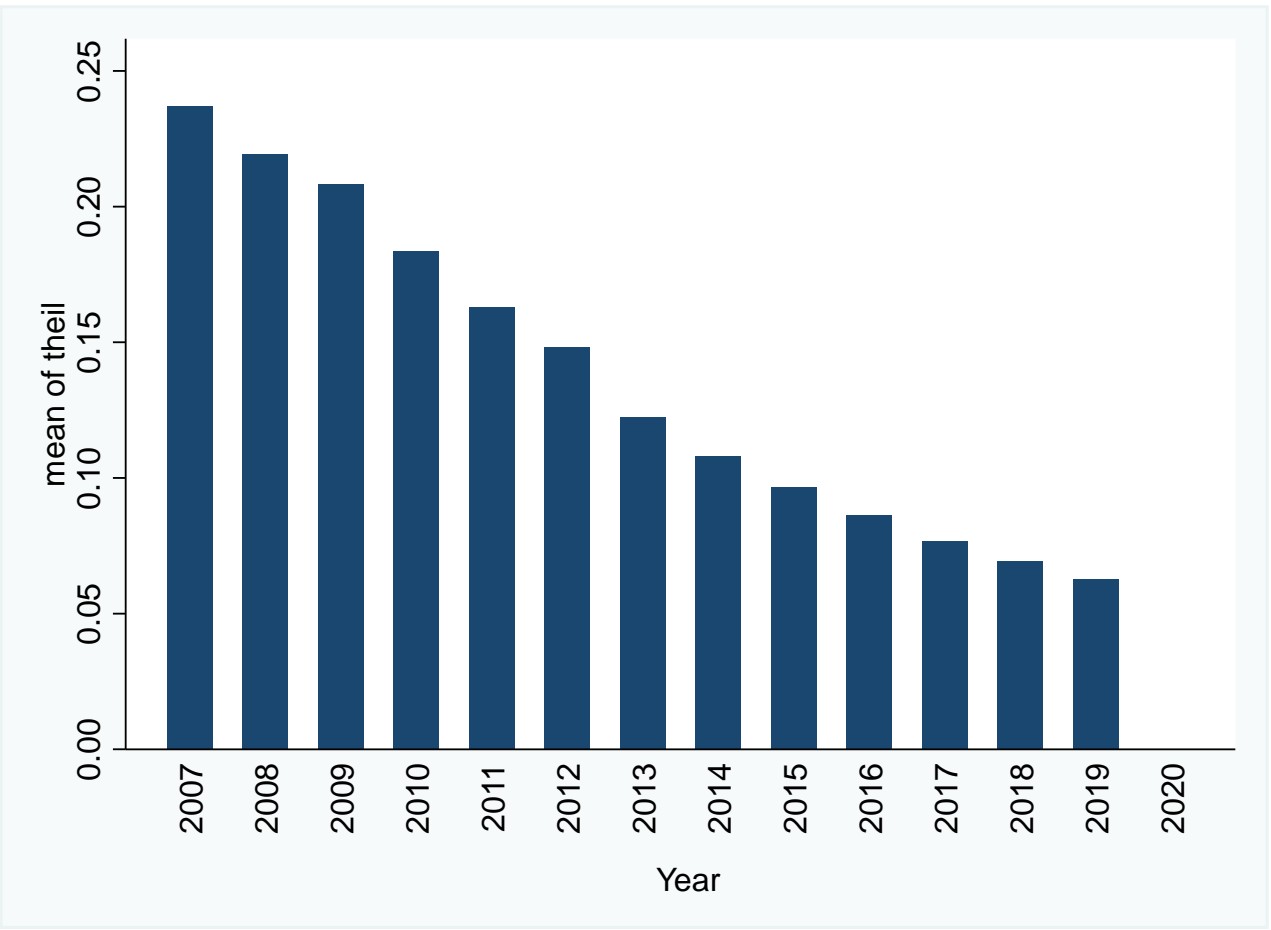

**Figure 2.** The average urban–rural income gap in China from 2007 to 2019.

From Figure 2, it can be seen that the average urban–rural income gap calculated by the Theil index in China's provinces basically showed a continuous downward trend during the observation period. The downward trend is due to the combined effects of economic, political, and other factors in China's entire society. The impact of the intelligentization of the circulation industry on the income gap between urban and rural areas in China needs to be further explored.

In order to better compare the changes in the urban–rural income gap between various regions of China over the past 13 years, the four graphs included in Figure 3 show the specific situation of the urban–rural income gap by province in different years. Figure 3A plots the comparison of urban and rural income gaps between China's provinces in 2007. From this map, we know that in 2007, the urban–rural income gap between Beijing, Shanghai and Guangdong was the largest, reaching over 0.065. In addition, the urban–rural income gap in Zhejiang Province is relatively large, above 0.057. Overall, the urban–rural income gap in Eastern China is the largest, and the urban–rural income gap in the northwest is the smallest.

Figure 3B shows a comparison of urban and rural income gaps between China's provinces in 2013. After the global financial crisis in 2008, the Chinese government issued subsidies of up to CNY 4 trillion to ease the economic pressure. In 2013, China's economy gradually recovered, and residents' incomes gradually returned to their levels before the economic crisis. However, the urban–rural income gap has increased across China. Compared with 2007, with the exception of Beijing, Shanghai, and Guangdong, the urban–rural income gap in Sichuan has risen to the first echelon. In addition, in Northeast China, the urban–rural income gap in Heilongjiang is smaller than in Jilin and Liaoning.

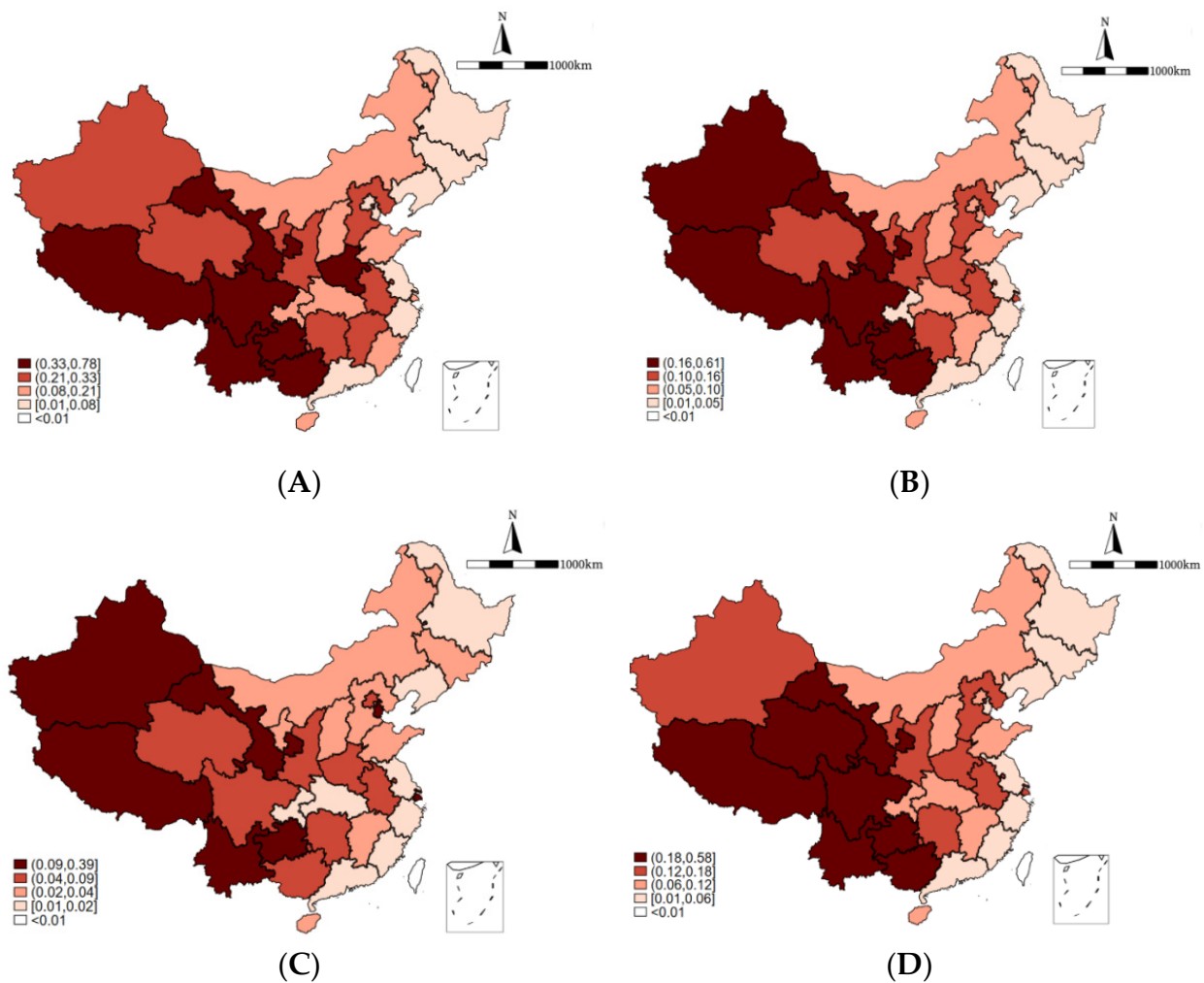

**Figure 3.** The urban–rural income gap maps: (**A**) 2007, (**B**) 2013, (**C**) 2019, and (**D**) means.

Figure 3C shows a comparison of urban and rural income gaps between China's provinces in 2019. Compared with 2013, the income gap between urban and rural areas in China's provinces has increased, and the provinces in the first tier remain unchanged. However, in the Northeast, the urban–rural income gap in Liaoning is larger than in the other two provinces.

Figure 3D shows a comparison of the average urban–rural income gap between China's provinces from 2007 to 2019. Beijing, Shanghai, Guangdong, and Sichuan provinces in China have the largest urban–rural income gap. The urban–rural income gaps in Liaoning and Zhejiang provinces are low, but are still at a high level compared to other provinces. Overall, the income gap between urban and rural areas in Southeastern China is relatively large, which may be caused by the development of the tertiary industry in the southeastern region. The urban–rural income gap in the northwest region is relatively small.

### 3.2. Spatial Analysis of the Circulation Industry in China

Figure 4 shows the changes in the intelligent circulation industry index in China between different years. Figure 4A plots a comparison of the intelligence index of the circulation industry in China's provinces in 2007. This map shows that Beijing, Tianjin, Henan, Hebei, Shandong, Jiangsu, Sichuan, Shanghai, and Guangdong had the highest level of intelligence in the circulation industry in 2007. Overall, the level of intelligent circulation industry in other regions is higher than that in the northern region in Southern China, except Guizhou and Chongqing. Figure 4B shows a comparison of the intelligent circulation industry index in China's provinces in 2013. Compared with 2007, the overall improvement

was higher, but the gap did not change much. Figure 4C shows a comparison of the intelligent circulation industry index in China's provinces in 2019. Compared with 2013, the intelligence level of the circulation industry in China's provinces has generally improved. However, in the Northeast region, the level of intelligence of the circulation industry in Liaoning Province is higher than that of the other two provinces. Figure 4D shows a comparison of the average value of the circulation-industry intelligence index in China's provinces from 2007 to 2019. Except for Guizhou and Chongqing, the degree of integration of the circulation industry and informatization in Southern China is stronger than that in northern China. In northern China, the intelligent development of the circulation industry in the northeast region is stronger than that in the northwest region.

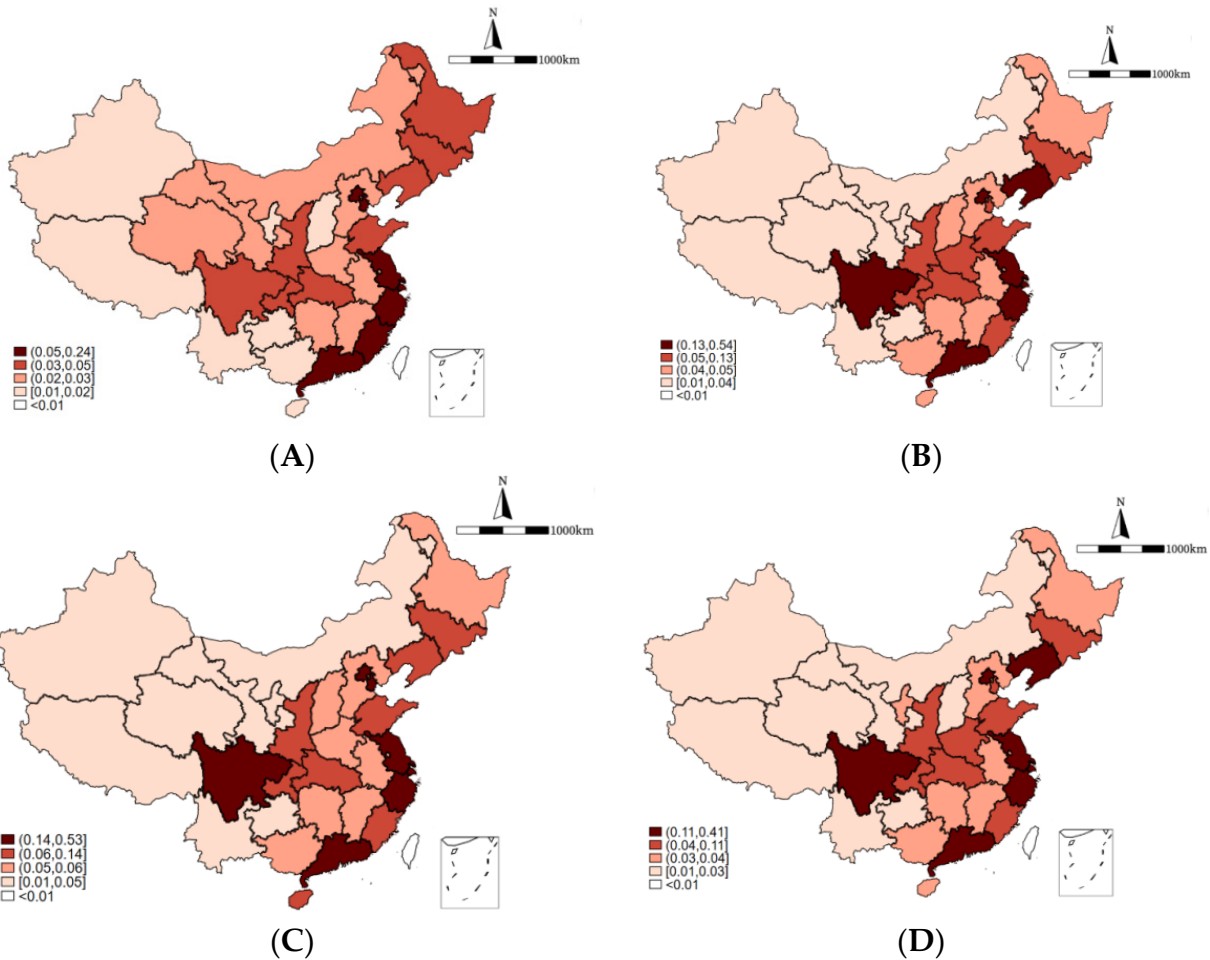

**Figure 4.** Maps of the intelligence level of the circulation industry: (**A**) 2007, (**B**) 2013, (**C**) 2019, and (**D**) means.

### 3.3. Temporal Analysis

In order to better show the trend in the intelligent development of the circulation industry in each province and the change in the urban–rural income gap, Figure 5 shows the changes in the two variables in each region from 2007 to 2019 across China.

As can be seen from Figure 5, the urban–rural income gap in China shows a downward trend. Gansu Province, Guangxi Province, Guizhou Province, Tibet, and Yunnan Province decline significantly. In addition, the speed of the intelligent circulation industry in China is different. The intelligent development of the circulation industry in Beijing first rises, and then, declines, and the intelligent development of the circulation industry in Guangdong Province, Sichuan Province, and Shanghai also fluctuate. Other provinces show a steady upward trend.

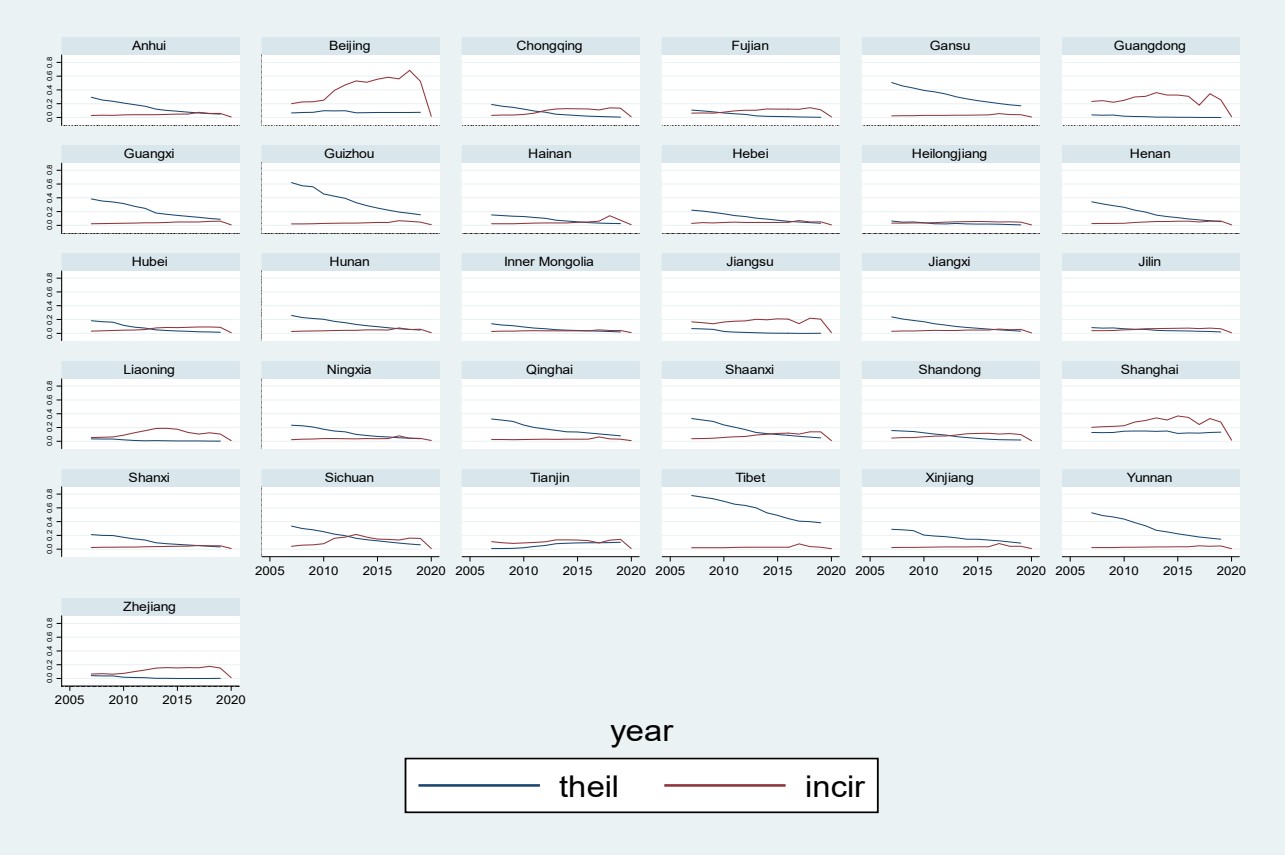

**Figure 5.** The trend in the urban–rural income gap and the intelligent circulation industry.

The results in Figure 6 reflect the comparative results of the intelligence of the circulation industry and the urban–rural income gap in China more clearly. This section uses the difference between 2007 and 2019 for each province to represent the speed at which the urban–rural income gap is reduced and the speed of the intelligent development of the circulation industry. From Figure 6A, it can be seen that the urban–rural income gap in the western region of China is decreasing faster than that in the eastern region. From Figure 6B, it can be seen that, except for Beijing and Zhejiang Province, the intelligent growth rate of the circulation industry in Eastern China is higher than that in other regions.

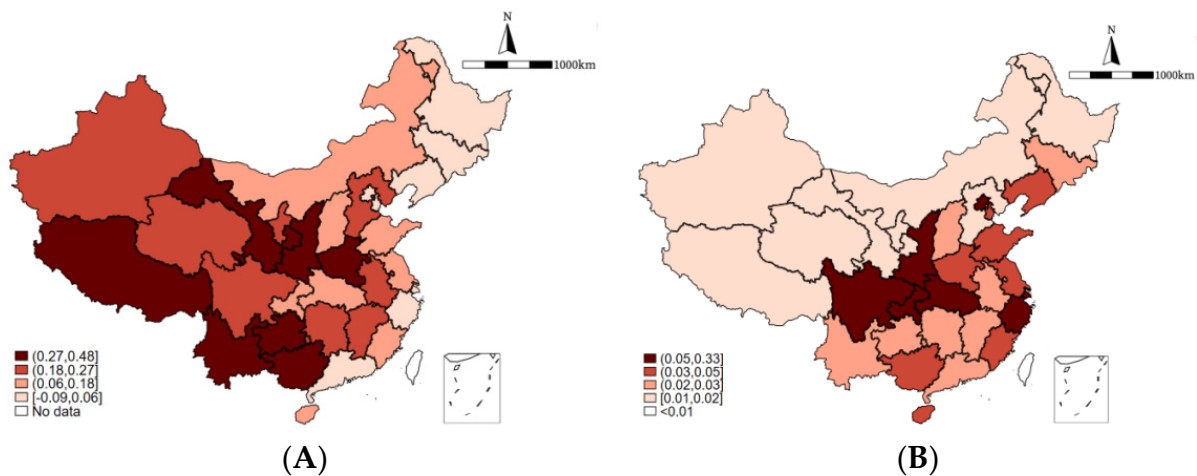

**Figure 6.** The trend in (**A**) the urban–rural income gap and (**B**) the intelligent circulation industry.

Table 3 and Figure 7 show the descriptive statistics of all the variables involved in the benchmark regression equation in this paper.

**Table 3.** Descriptive statistics results.

| Variable | Obs | Mean | Std. Dev. | Min | Max |
|---|---|---|---|---|---|
| *theil* | 403 | 0.136993 | 0.139331 | 1.36E-05 | 0.778484 |
| *incir* | 434 | 0.085779 | 0.09681 | 0.00732 | 0.684032 |
| *ri* | 372 | 101.6051 | 60.6724 | 2.361749 | 324.1569 |
| *lgdp* | 434 | 4.692387 | 0.031048 | 4.553877 | 4.770685 |
| *pk* | 403 | 11.93212 | 26.68949 | 0.233645 | 226.7098 |
| *lexgdp* | 434 | 7.884797 | 0.974196 | 4.706965 | 10.04322 |
| *lingdp* | 434 | 5.615799 | 0.840386 | 1.800528 | 8.345198 |
| *innogdp* | 434 | 0.004385 | 0.002519 | 0.001433 | 0.013677 |

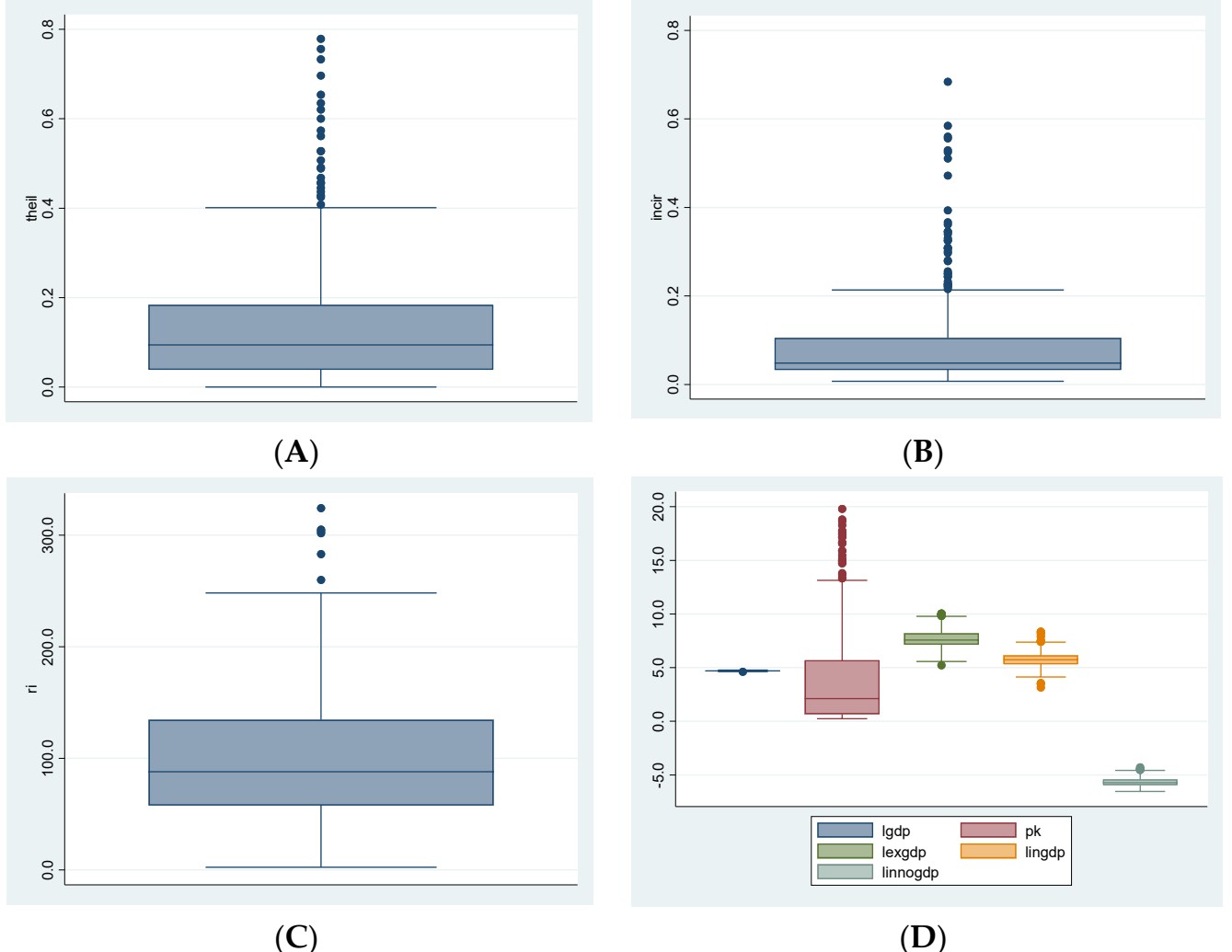

**Figure 7.** Boxplot of variables. (**A**) Boxplot of *theil*; (**B**) Boxplot of *incir*; (**C**) Boxplot of *ri*; (**D**) Boxplot of control variables.

In order to analyze the distribution of variables in different regions more intuitively, our research divides China into three parts: eastern, central and western by geographical location. The obtained results are shown in Figure 8.

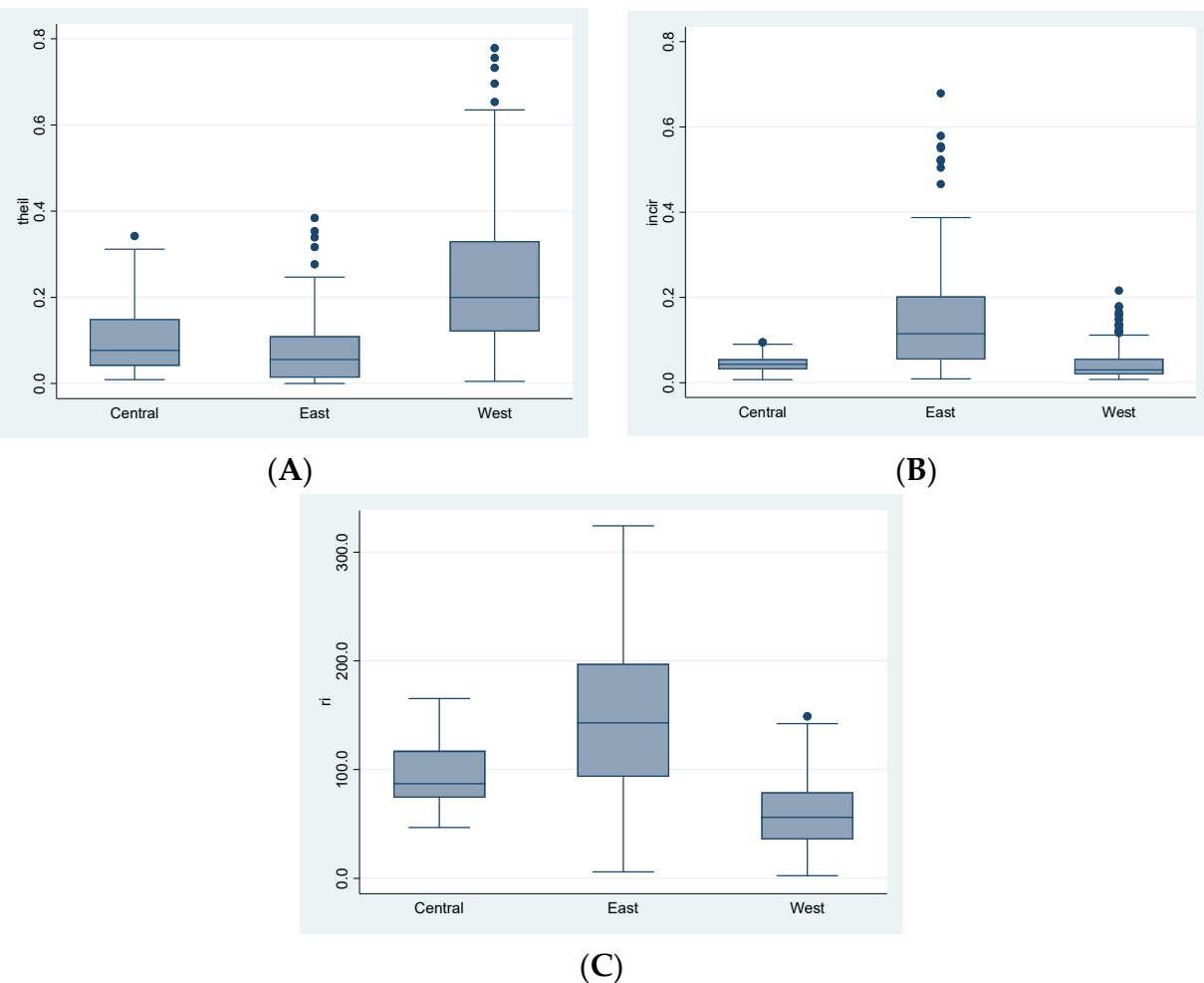

**Figure 8.** Boxplots of main variables by area. (**A**) Boxplot of *theil* by area; (**B**) Boxplot of *incir* by area; (**C**) Boxplot of *ri* by area.

From Figure 8A, it can be seen that the overall level of the urban–rural income gap in Western China is the highest and the span is the largest. The overall level of the urban–rural income gap in Eastern China is the lowest. From Figure 8B, it can be seen that the circulation industry in Eastern China has the highest level of intelligence with the largest span, which shows that there is a large gap in the speed of integration between the circulation industry and the information industry in the eastern region. The circulation industry in Western China has the lowest level of intelligence. Central China's circulation industry has the smallest span of intelligence. From Figure 8C, it can be seen that the comparison results of the ratio of innovation investment of GDP in various regions of China are similar to those of the level of intelligence of the circulation industry.

### 3.4. Main Model Regression Results

All the regressions in this research used a fixed-effects model, and the results of the benchmark regression equation are shown in Table 4. Models (1)–(3) are the model results without control variables, and models (4)–(6) are the model results with control variables. Models (1) and (4) show the effect of the intelligence of the circulation industry on the average annual disposable income of households in China's urban areas. Models (2) and (5) show the effect of the intelligence of the circulation industry on China's rural areas and its impact on the annual disposable income per person of regional households. Models (3) and (6) show the effect of the intelligence of the circulation industry on the income gap between urban and rural residents in China.

**Table 4.** Benchmark regression results.

|  | Model (1) | Model (2) | Model (3) | Model (4) | Model (5) | Model (6) |
|---|---|---|---|---|---|---|
|  | *cityincome* | *countryincome* | *theil* | *cityincome* | *countryincome* | *theil* |
| *incir* | 36,324.3 *** | 13,628.0 *** | 0.842 *** | 43,864.3 *** | 11,960.2 *** | 0.541 *** |
|  | (3.71) | (3.10) | (3.10) | (7.78) | (4.94) | (3.88) |
| *incir*2 |  |  | −0.949 ** |  |  | −0.477 ** |
|  |  |  | (−2.44) |  |  | (−2.47) |
| *lgdp* |  |  |  | −203,810.1 *** | −95,727.5 *** | −0.538 *** |
|  |  |  |  | (−21.55) | (−23.59) | (−4.69) |
| *pk* |  |  |  | 143.2 *** | 74.80 *** | −0.000442 *** |
|  |  |  |  | (13.30) | (16.19) | (−3.84) |
| *lexgdp* |  |  |  | −2975.2 *** | −720.2 ** | −0.0225 *** |
|  |  |  |  | (-4.49) | (−2.53) | (-3.09) |
| *lingdp* |  |  |  | 125.9 | −273.1 | 0.0379 *** |
|  |  |  |  | (0.26) | (−1.29) | (7.16) |
| *innogdp* |  |  |  | 491,317.5 *** | 299,127.8 *** | 1.141 |
|  |  |  |  | (2.69) | (3.81) | (0.58) |
| _cons | 23,810.8 *** | 9298. 1*** | −1.354 *** | 998,005.0 *** | 463,482.4 *** | 1.158 ** |
|  | (28.34) | (24.65) | (−73.88) | (22.85) | (24.74) | (2.19) |
| N | 434 | 434 | 403 | 403 | 403 | 403 |
| $R^2$ | 0.032 | 0.022 | 0.125 | 0.856 | 0.871 | 0.327 |

*t* statistics in parentheses. ** $p < 0.05$, *** $p < 0.01$.

In models (1) and (2), the coefficients of *incir* represent the degree of intelligence of the circulation industry, which are both positive and significant at a 1% confidence level. By comparison, it can be seen that the coefficient of variable *incir* in model (1) is higher than the coefficient of variable *incir* in model (2) without controlling for economic development, e-commerce development, foreign trade, traditional transportation, and information-technology development. This result shows that under the condition of not controlling external conditions, the promotion effect of the intelligent circulation industry on the income of urban residents is higher than on that of rural residents. Similarly, after adding control variables, the coefficients of the variable *inir* in models (4) and (5) are also positive, and both are significant at a 1% confidence level. Additionally, the coefficient of variable *incir* in model (4) is higher than the coefficient of variable *inir* in model (5). This result shows that after controlling for external variables, the promotion effect of the intelligent development of the circulation industry on the income of urban residents is still higher than on that of rural residents.

Similarly, through comparison of models (4) and (5), it can be seen that the intelligent development of the circulation industry has different promotion effects on the income of urban residents and rural residents. The promotion effect of the intelligent circulation industry on the income of urban residents is higher than on that of rural residents. In order to further investigate the influence of the intelligent development of the circulation industry on the urban–rural income gap, the secondary term of the intelligent development of the circulation industry is added for analysis. With no control variables, the quadratic coefficient in model (3) is negative and significant at a 5% confidence level. Furthermore, the coefficient of the variable *inir* is positive and significant at a 1% confidence level. The results of these two aspects show that the intelligent development of the circulation industry has an inverted "U" shape on the income gap between urban and rural areas in China. Before reaching the threshold, the agglomeration effect caused by the intelligent development of the circulation industry is greater than the diffusion effect, so the intelligent development of the circulation industry will increase the income gap between urban and rural areas. After reaching the threshold, the diffusion effect caused by the intelligent development of the circulation industry is greater than the agglomeration effect, so the intelligent development of the circulation industry will narrow the income gap between urban and rural areas. Model (6) exhibits similar results to model (3) after adding control variables. The quadratic coefficients in model (6) are still negative and significant at a 5% confidence level. The coefficient of the variable inir remains positive and significant at a 1% confidence level. This

result verifies Hypothesis 1, explaining the inverted "U"-shaped impact of the intelligent development of the circulation industry on the urban–rural income gap.

Figure 9 shows that with improvement in the intelligence level of the circulation industry, the income gap between urban and rural areas in China first increases, and then, decreases.

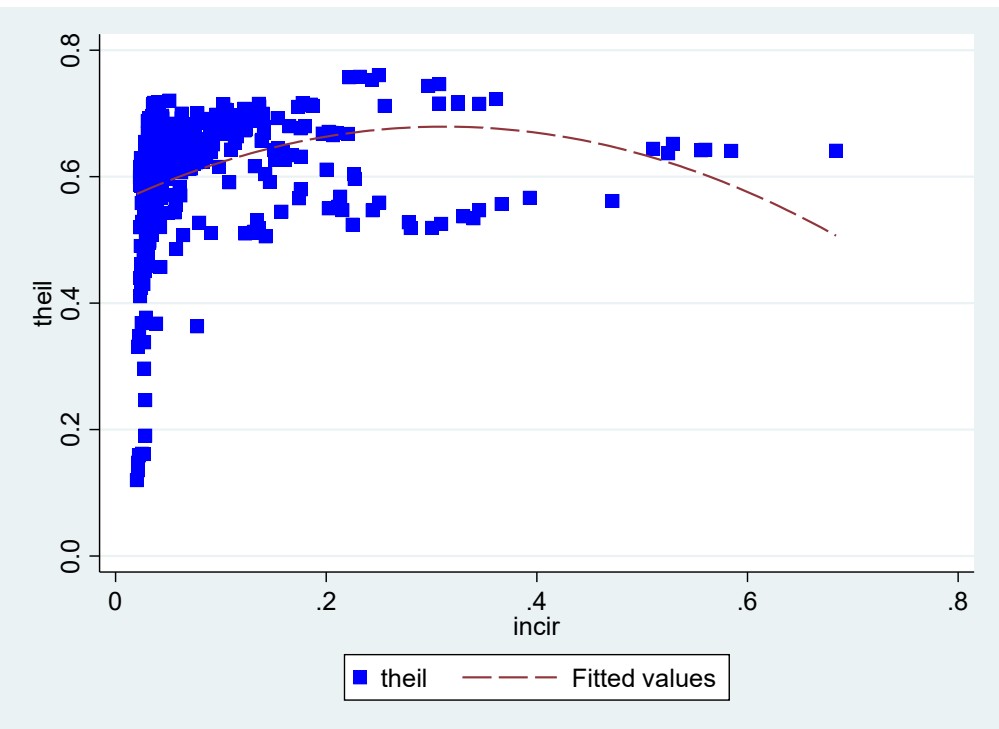

**Figure 9.** The relationship between the intelligent circulation industry and the urban–rural income gap.

### 3.5. Test of Mediation Effect

After verifying that the intelligent circulation industry has an inverted "U"-shaped impact on the urban–rural income gap, this research further explored the intermediary mechanism that causes this. From the theoretical analysis, it can be considered that the intelligent circulation industry will affect local innovation investment. The ratio of innovation and transformation is higher in regions with a high degree of intelligent circulation industry, which also affects the urban–rural income gap.

In order to verify the inverted "U"-shaped impact of the proportion of innovation investment on the intelligence of the circulation industry, we return to Equations (5) and (6). Models (7) and (9) use the proportion of innovation investment with the mediation effect as the outcome variable, and show the regression results of Equation (5). Among them, model (7) does not add control variables, and model (9) shows the results with control variables. Models (8) and (10) use the explained variable $theil_{it}$ as the outcome variable, and show the regression results of Equation (6). Among them, model (8) does not add control variables, and model (10) adds control variables.

It can be seen from Table 5 that the intelligent circulation industry has a significant impact on the proportion of innovation investment. First, we analyze the regression results with no explanatory variables. The coefficient of the incir variable representing the intelligent circulation industry in model (7) is positive and significant at a 5% confidence level. At the same time, the quadratic term coefficient in this model is negative. It is verified that with the intelligent development of the circulation industry, the proportion of innovation investment first increases, and then, decreases. Model (8) verifies the positive correlation between the proportion of innovation investment and the urban–rural income gap, and the

result is also significant at a 10% confidence level. These two models, combined with the results of model (3), verify that innovation investment has a significant mediating effect regarding the intelligence of the circulation industry on the urban–rural income gap. Statistically, it can be considered that with the intelligent development of the circulation industry, the proportion of regional innovation investment first increases, and then, decreases, thus driving the local urban–rural income gap to increase first, and then, decrease.

**Table 5.** Regression results of mediation effect.

|  | **Model (7)** | **Model (8)** | **Model (9)** | **Model (10)** |
|---|---|---|---|---|
|  | *ri* | *theil* | *ri* | *theil* |
| *incir* | 125.2 ** | | 395.1 *** | |
|  | (2.06) | | (6.41) | |
| *incir*2 | −140.5 | | −480.4 *** | |
|  | (−1.60) | | (−5.48) | |
| *ri* | | 0.000434 * | | 0.000269 ** |
|  | | (1.91) | | (2.22) |
| *lgdp* | | | 15.70 | −0.703 *** |
|  | | | (0.28) | (−6.35) |
| *pk* | | | −0.00419 | −0.000261 ** |
|  | | | (−0.08) | (−2.41) |
| *lexgdp* | | | 4.023 | −0.0244 *** |
|  | | | (1.23) | (−3.37) |
| *lingdp* | | | 2.530 | 0.0353 *** |
|  | | | (1.01) | (6.29) |
| *innogdp* | | | 8208.0 *** | −0.638 |
|  | | | (9.46) | (−0.29) |
| _cons | 92.94 *** | −1.331 *** | −84.47 | 1.984 *** |
|  | (24.01) | (−58.18) | (−0.33) | (3.95) |
| *N* | 372 | 341 | 341 | 341 |
| $R^2$ | 0.029 | 0.046 | 0.438 | 0.298 |

*t* statistics in parentheses. * $p < 0.1$, ** $p < 0.05$, *** $p < 0.01$.

After adding control variables, the mediating effect is more significant. The results of model (9) show that after controlling for factors such as social economy, transportation, and technology, the degree of intelligence of the circulation industry has an inverted "U" shape on the proportion of innovation investment. Additionally, the coefficients of both primary and secondary terms are significant at a 1% confidence level. Model (10) further verifies that the proportion of innovation investment still has a positive impact on the urban–rural income gap after controlling for other factors, and the impact is significant at a 5% confidence level. Combined with the results of model (6), it can be considered that after adding control variables, the mediating effect of the proportion of innovation investment is still significant. The results of models (7) and (9) verify Hypothesis 2, indicating that the impact of the intelligent development of the circulation industry on the proportion of innovation investment is also in an inverted "U" shape. The results of models (8) and (10) verify Hypothesis 3, indicating that the proportion of innovation investment and the urban–rural income gap change in the same direction. The above results prove that the proportion of innovation investment is an important transmission mechanism in the process of the intelligent development of the circulation industry and affects the urban–rural income gap.

The plot in Figure 10 exhibits an inverted "U" shape. This result shows that with improvement in the intelligence level of the circulation industry, the proportion of investment first increases, and then, decreases, which is consistent with the results of the model.

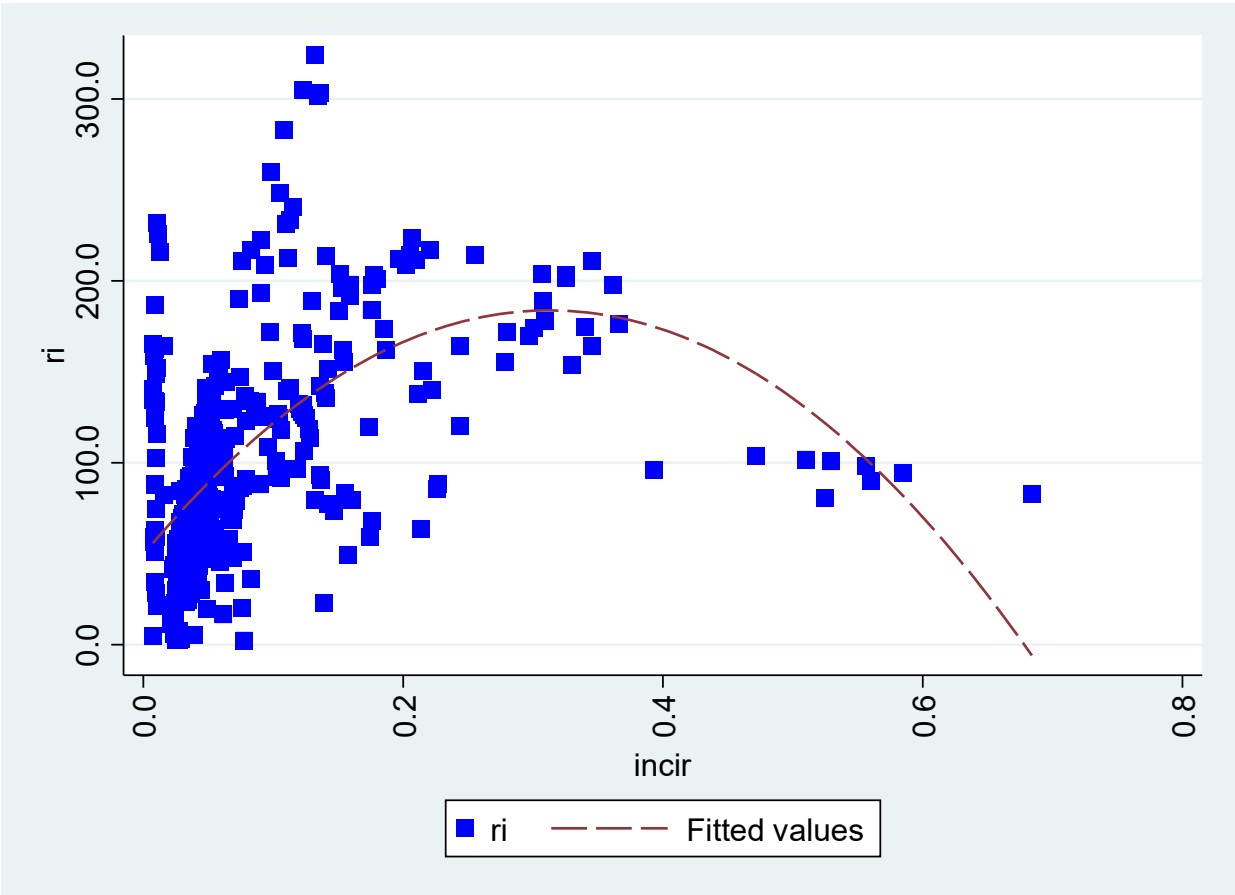

**Figure 10.** The relationship of the intelligent circulation industry and proportion of innovation investment.

### 3.6. Endogeneity and Robustness

#### 3.6.1. Endogenous Test

In the regression, there may exist omitted variables, mutual causality, and other factors that will cause endogenous bias. In order to verify the accuracy of the results, this research constructed a dynamic panel model and used the systematic GMM method of regression. In the benchmark regression model (11), we added the three lag periods of the urban–rural income gap as explanatory variables, and the one lag period of the intelligentization of the circulation industry as an instrumental variable. In the mediation-effect verification model (12), the lag period of the proportion of innovation investment was used as an explanatory variable, and the lag period of circulation-industry intelligence was still used as an instrumental variable. Other control variables remained unchanged. The final results are shown in Table 6.

From the results obtained in Table 6, it can be seen that whether the urban–rural income gap is used as the outcome variable in the benchmark regression, and whether the ratio of innovation investment is used as the outcome variable in the mediation-effect test regression, the coefficients of the final linear coefficient inir are all positive, and the coefficients of the quadratic term coefficient inir2 are all negative; additionally, all the explanatory variables are significant at the 1% confidence level. This result shows that the intelligent circulation industry has an inverted "U"-shaped impact on both the urban–rural income gap and the proportion of innovation investment. In summary, the stability of the conclusions of this research is further verified by the regression results of the GMM method.

In addition, it can be seen in model (11) that both the first-order lag term and the second-order lag term of the urban–rural income gap will have a positive impact on the current urban–rural income gap, but the third-order lag term of the urban–rural income gap will have a positive impact on the current urban–rural income gap, not a negative

impact. This may be because the local policy-issuing agency will formulate corresponding improvement policies according to the urban–rural income gap in the region, but the impact on the policy has a time lag. Correspondingly, the first-order lag term of the proportion of innovation investment in model (12) also has a positive impact on the proportion of innovation investment in the current period, which is related to the long-term impact on the investment of regional development.

**Table 6.** System GMM regression results.

|  | **Model (11)** | **Model (12)** |
|---|---|---|
|  | *theil* | *ri* |
| *incir* | 0.0869 *** | 53.45 *** |
|  | (9.35) | (3.78) |
| *incir*2 | −0.0323 *** | −49.17 *** |
|  | (−3.17) | (−3.00) |
| *lgdp* | −0.151 *** | 87.12 *** |
|  | (−9.01) | (6.86) |
| *pk* | −0.000131 *** | −0.441 *** |
|  | (−7.57) | (−14.52) |
| *lexgdp* | −0.00343 *** | −20.51 *** |
|  | (−4.23) | (−6.59) |
| *lingdp* | 0.00659 *** | 3.873 *** |
|  | (18.87) | (14.71) |
| *innogdp* | 0.838 *** | 7643.8 *** |
|  | (2.72) | (11.60) |
| L.*theil* | 0.818 *** |  |
|  | (68.22) |  |
| L2.*theil* | 0.229 *** |  |
|  | (17.71) |  |
| L3.*theil* | −0.243 *** |  |
|  | (−26.88) |  |
| L.*ri* |  | 0.552 *** |
|  |  | (27.91) |
| *N* | 279 | 217 |

*t* statistics in parentheses. *** $p < 0.01$.

## 3.6.2. Robustness Test

The robustness test can be carried out by replacing the explained variables. Referring to the method of Li et al. (2020), we used the ratio rate$_{it}$ of the per capita disposable income of urban households to that of rural households as a surrogate variable for the urban–rural income gap to conduct the test.

Models (13) and (15) show the influence of the degree of intelligent circulation industry on the urban–rural income gap without and with control variables, respectively. From the results in Table 7, it can be seen that at a confidence level of 1%, the degree of intelligence of the circulation industry still has an inverted "U"-shaped impact on the urban–rural income gap after the substitution of variables. Similarly, models (14) and (16) show the impact on the proportion of innovation investment of the urban–rural income gap after the variable is replaced with and without the control variable, respectively. The results in Table 7 show that the degree of intelligent circulation industry has a positive impact on the urban–rural income gap after the substitution variable, which is significant at a 1% confidence level. The above results verify the stability of the conclusions, indicating that the proportion of innovation investment is, indeed, an intermediary mechanism for the inverse "U"-shaped impact on the intelligentization of the circulation industry on the urban–rural income gap. Table 7 presents the results of the robustness test.

**Table 7.** Robustness test results.

|  | **Model (13)** | **Model (14)** | **Model (15)** | **Model (16)** |
|---|---|---|---|---|
|  | *rate* | *rate* | *rate* | *rate* |
| *incir* | 0.343 *** | | 0.523 *** | |
|  | (3.94) | | (7.52) | |
| *incir*2 | −0.567 *** | | −0.920 *** | |
|  | (−4.89) | | (−9.55) | |
| *ri* | | 0.000792 *** | | 0.000420 *** |
|  | | (3.68) | | (6.26) |
| *lgdp* | | | −0.721 *** | −0.949 *** |
|  | | | (−12.62) | (−15.54) |
| *pk* | | | 0.0000839 | 0.0000638 |
|  | | | (1.47) | (1.07) |
| *lexgdp* | | | 0.0120 *** | 0.0167 *** |
|  | | | (3.31) | (4.18) |
| *lingdp* | | | 0.00469 * | 0.00624 ** |
|  | | | (1.78) | (2.02) |
| *innogdp* | | | 5.486 *** | 1.589 |
|  | | | (5.65) | (1.30) |
| _cons | 0.353 *** | 0.299 *** | 3.580 *** | 4.608 *** |
|  | (63.71) | (13.67) | (13.64) | (16.65) |
| *N* | 434 | 372 | 403 | 341 |
| $R^2$ | 0.059 | 0.261 | 0.696 | 0.635 |

*t* statistics in parentheses. * $p < 0.1$, ** $p < 0.05$, *** $p < 0.01$.

### 3.7. Regional Heterogeneity

Due to the large land area of China and the different economic development foundations of different regions, China was divided into eastern, central, and western regions according to geographical location; this enabled us to conduct sub-regional regression tests and examine the differences in the intelligent development of the circulation industry in different economic development regions. The results are shown in Table 8.

**Table 8.** Regional Heterogeneity Analysis.

|  | **Model (17)** | **Model (18)** | **Model (19)** |
|---|---|---|---|
|  | East | Central | West |
| *incir* | 0.117 | 6.575 *** | 2.850 *** |
|  | (0.81) | (7.52) | (5.43) |
| *incir*2 | 0.0873 | −53.37 *** | −10.52 *** |
|  | (0.46) | (−7.53) | (−4.46) |
| *lgdp* | −0.0609 | −0.0443 | −0.225 |
|  | (−0.40) | (−0.39) | (−0.81) |
| *pk* | 0.0000292 | 0.00237 *** | 0.00253 * |
|  | (0.27) | (4.14) | (1.76) |
| *lexgdp* | 0.0132 | 0.0233 *** | −0.0419 *** |
|  | (0.85) | (2.75) | (−4.18) |
| *lingdp* | 0.0195 *** | 0.0131 ** | 0.0716 *** |
|  | (3.24) | (2.37) | (6.01) |
| *innogdp* | −7.919 *** | 2.753 | −0.981 |
|  | (−3.37) | (1.31) | (−0.22) |
| _cons | −1.176 * | −1.517 *** | −0.551 |
|  | (−1.72) | (−2.87) | (−0.42) |
| *N* | 156 | 117 | 130 |
| $R^2$ | 0.191 | 0.771 | 0.717 |

*t* statistics in parentheses. * $p < 0.1$, ** $p < 0.05$, *** $p < 0.01$.

According to the results obtained in Table 8, it can be seen that the impact of the development of the intelligentization circulation industry on the urban–rural income gap is

quite different in China. The results of models (18) and (19) show that the inverted "U"-shaped impact of intelligentization circulation-industry development on the urban–rural income gap is significant in Central and Western China, but the effect is not significant in Eastern China in model (17). This result may be due to the high degree of marketization, the faster development of informatization, and the more prominent characteristics of uneven industry penetration in Eastern China, which have widened the regional income gap and the urban–rural income gap.

## 4. Discussion

The arrival of the digital-economy era marked an improvement in the degree of informatization. Further, narrowing the urban–rural income gap can effectively promote the coordinated development of urban and rural areas. This is an important step in promoting sustainable economic development, and is still of key significance in this digital-economy era. Furthermore, the development of informatization provides new ideas for the development of the economy in rural areas, with limited infrastructure. The development of circulation-industry intelligence has also broadened employment channels in rural areas and improved social productivity. Thus, by balancing the speed of economic development between urban and rural areas, circulation-industry intelligence contributes to regional coordination and sustainable development.

Unlike the traditional analyses of the circulation industry, our study introduces the concept of circulation-industry intelligentization; based on the theory of agglomeration and diffusion effects, in this study, we constructed a fixed-effects model to analyze the impact of the intelligent development of the circulation industry on the urban–rural income gap. Further, we also verified whether innovation investment is a significant mediating mechanism in the impact process.

On a theoretical basis, and based on empirical analysis, our results showed that the impact of the development of the intelligence of China's circulation industry on the urban–rural income gap was non-linear, and overall, showed an inverted "U"-shaped relationship. Based on the analysis, we further observed that at the early stage of the integration of the circulation industry with the information industry, urban areas with more capital and labor endowments developed faster, implying that innovation investments in these areas resulted in greater benefits. Therefore, the agglomeration effect was significantly higher than the diffusion effect. Consequently, the incomes of urban residents increased accordingly, leading to an increase in the urban–rural income gap. However, with the development of circulation-industry intelligence at later stages, land, rent, and other costs in urban areas increased; thus, the diffusion effect gradually increased. This implies that innovation investment could be transferred to rural areas at a lower cost. Therefore, accelerating the development of circulation-industry intelligence can improve overall development as well as income in rural areas, thereby narrowing the urban–rural income gap.

To balance the income gap between urban and rural areas and promote the sustainable development of China's economy, it is necessary to balance innovation investment in both urban and rural areas; moreover, it is necessary to reduce the differences between these two areas with respect to the level of intelligence of the urban and rural circulation industries. Notably, in this study, we observed that this effect was regionally heterogeneous. Our analysis showed that in areas with a lower degree of marketization and a slower rate of informatization, industry penetration was more balanced, and the non-linear impact of the intelligentization of the circulation industry on the urban–rural income gap was more apparent. Further, considering the international theme of sustainable development, the intelligentization of the circulation industry could effectively reduce carbon emissions and help China achieve regional environmental policy goals. These findings have important practical significance and provide an effective reference for other countries and regions pursuing sustainable and coordinated development.

In the context of the digital economy, the development trend in the circulation industry is to integrate informatization. However, it is not possible to ignore the fact that the

collision between economy and efficiency inevitably brings about certain moral challenges to social development [77,78]. For example, although the advancement of technologies, such as autonomous driving, has important practical significance for the development of transportation efficiency in the case of an epidemic, the popularization of informatization has enhanced the challenges associated with transportation. People who are reluctant to accept new information would not proactively accept an intelligent traffic environment. In other words, the development of informatization results in certain user groups, comprising people who find it hard to accept new technologies, being left out. These factors are also the key points that need to be considered during the process of circulation-industry intelligence development.

## 5. Conclusions

In the context of the information age, industrial intelligence refers to development that seeks to integrate informatization into different industries, including the circulation industry. In this study, we observed that technological progress in the circulation industry promotes the narrowing of the urban–rural income gap through channels such as reduced transportation costs, accelerated labor mobility, the availability of ideas regarding regional economic coordination, and sustainable development.

The income gap between urban and rural areas in China has been continuously decreasing, and the level of circulation-industry intelligence has been continuously improving. However, all of these variations showed regional heterogeneity. Based on the results related to the agglomeration and diffusion effects, the development of circulation-industry intelligence displayed a staged and differentiated impact on the urban–rural income gap, and also showed an overall inverted "U"-shaped trend. We also observed that innovation investment is a significant intermediary mechanism in the process of circulation-industry intelligentization, exerting a narrowing effect on the urban–rural income gap.

Further, based on theoretical analysis and empirical research, in this study, we systematically discussed the nonlinear impact of the integration of informatization into the circulation industry on the income gap between urban and rural residents; this provides a basis and reference for other countries and regions in the world that seek to pursue coordinated regional economic development. Therefore, considering our obtained results, the following recommendations can be made: first, during the development of circulation-industry intelligence, it is necessary to consider the issue of urban and rural income gaps as a whole; second, with the process of urbanization, it important to also speed up the development of rural informatization; third, countries need to increase employment opportunities, reduce taxes, or provide subsidies for new industries in rural areas so as to attract the return of labor to these areas. Additionally, given that innovation investment is a significant intermediary mechanism, it is recommended that governments should take rural areas into consideration when making regional innovation investments. Moreover, in terms of FDI, it is necessary for governments and other management agencies to guide enterprises to invest in rural areas so as to balance the rate of circulation-industry intelligence development and narrow the income gap between urban and rural areas, while promoting the sustainable development of the regional economy.

This study has some limitations. First, this study used the fixed-effects method for testing, which does not take into account the interaction between regions. Second, we only included data corresponding to the 2007–2019 period. This could be considered less reliable given that after 2019, the effects of several uncontrollable factors such as the COVID-19 pandemic and international wars on the international economy have increased. All of these exogenous factors could have had an impact on the quality of the results obtained. Therefore, there is still room for improvement in the analyses of future studies.

**Author Contributions:** Conceptualization, methodology, software, validation, formal analysis, investigation, resources, writing—original draft preparation, visualization, supervision, H.M.; data curation, H.M. and P.D.; writing—review and editing, H.M., P.D. and J.Z. All authors have read and agreed to the published version of the manuscript.

**Funding:** This research received no external funding.

**Conflicts of Interest:** The authors declare no conflict of interest.

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
