# Peer review of "Nonlinear Impact of Circulation-Industry Intelligentization on the Urban–Rural Income Gap: Evidence from China"

_sustainability, doi:10.3390/su14159405_

Round 1

Reviewer 1 Report

I find that this is a strong quantitative paper. However, my concern is about a more transparent discussion of Chinese data sources and a dialogue with other similar papers. In my view the authors use their recent works as substantive references.

Author Response

Thank you for your fair suggestion to our study. In response to your suggestion, we add more content related to China's urban-rural income gap in the part of the introduction and theoretical mechanism discussion to ensure the reliability of data selection.

In addition, compared with existing research, our study has reliable basis and innovation in the selection of topics and data.

First, about the topic selection. The topic of our study is the impact of the intelligentization of the circulation industry on the income gap between urban and rural areas. However, most of the existing literature examines the impact on the urban-rural income gap based on transportation infrastructure. About the research related to the combination of informatization and transportation, most of the existing papers analyze the content related to e-commerce or ICT, and there is no analysis of the overall situation of the circulation industry. In the introduction part, the innovation points of our study are analyzed and discussed detailed.

Second, about the data selection, our study is supported by many literature. On one hand, in the process of analyzing the impact of the intelligentization of the circulation industry on the urban-rural income gap, our study analyzes the channels that affect the urban-rural income gap from two aspects: traditional transportation infrastructure and informatization. On the other hand, based on the two theories of agglomeration effect and diffusion effect, our study analyzes the influence of the circulation industry, informatization and innovation investment on the urban-rural income gap. When selecting specific indicators, the construction of the circulation industry intelligence index refers to the research of Sun et al. (2021). The construction of the urban-rural income gap index is based on the research of Tang et al. (2022) and others, which are based on literature, and Tables 1 and 2 in the paper fully display the construction indicators of all variables, combined with the clear content of Research 2.2. Describes how all variables are constructed.

In conclusion, the data basis, source and innovation of our study are all reliable.

I look forward to working with the reviewers to move our manuscript closer to publication in Sustainability. The manuscript has been rechecked and the necessary changes have been made in accordance with the reviewers’ suggestions. The revised manuscript have been prepared and given below.

Reviewer 2 Report

Title:

1/ condensate the main revelation into a short and groundbreaking claim

Abstract:

2/ better follow the established schema of writing academic Abstract: A/ introduction (urgency and significance of the research hypothesis); B/ principles of the methods used + key results; C/ conclusions (commercial and environmental impacts)

3/ originality is questionable, clearly indicate how will our international audience of readers benefit from these findings

Introduction:

4/ go straight to the point and more in depth, write more technically (always provide corresponding numbers), significantly condensate all the text by reducing ballast phrases and cliché

5/ better address our international audience of readers, make sure the topic is not limited to any specific country

6/ the research hypothesis is not clearly stated, clearly justify the urgency and importance of its investigation, clearly identify those who will benefit from the findings

Materials and Methods:

7/ the method must be presented in such a way that it can be reproduced anytime, by anyone, anywhere (do not create obstacles like referring to specific location etc.)

Discussion:

8/ show more self-criticism to your work (can all the methods and results be fully trusted? what are the weaknesses of the methods used? where do the main measurement inaccuracies arise? what are the limitations from a commercial point of view? are the lessons learned transferable to other fields?)

9/ do not ignore the ethical aspects, refer to paper "Study on agriculture decision-makers behavior on sustainable energy utilization"

10/ reveal the main driving mechanisms of your results, provide deeper synthesis and reveal some more original/significant findings

Conclusions:

11/o not repeat your methods and results again and again, please understand that the Conclusion chapter is not a summary of your work, present only original and industrially significant revelations that have the potential to expand the horizon of human knowledge (higher level of generalization is mandatory)

12/ the originality is questionable, provide deeper synthesis

Author Response

Title:

1/ In order to highlight the relationship between explanatory variables and explained variables more clearly, the title of our study is changed to "The Nonlinear Impact of Intelligentization of Circulation Industry on Urban-Rural Income Gap: Evidence from China".

This title succinctly illustrates the relationship between the explanatory variable intelligence circulation industry and the outcome variable urban-rural income gap.

Abstract:

2, 3/ Based on your kind suggestion, we revise the abstract as follows:

The integration of the circulation industry with informatization has formed the concept of intelligence of the circulation industry. By reducing transportation costs and increasing total factor productivity, the income of residents can be improved in rural areas; a new pattern of regional economy can be shaped; urban, rural, social and economic development can be coordinated; and social sustainable development can be promoted. In this study, we used China's provincial panel data from 2007 to 2019 to measure the intelligence index of the circulation industry in each region and determine the indicators of the urban-rural income gap, and then conducted a comparative analysis. A fixed effect model was established based on the theory of the agglomeration and diffusion effect to analyze the relationship between the two variables. The study proved that innovation investment was a significant intermediary mechanism. The robustness of the results was ensured by substituting variables and controlling for endogeneity, and the effect was shown to be regionally heterogeneous. This study innovatively integrates informatization into the concept of the circulation industry, and the results provide a reference for formulating transportation infrastructure and informatization strategies to promote urban-rural coordination and sustainable development all around the world.

Introduction:

4/ According to your kind suggestion, we have added statistics on the urban-rural income gap in the first paragraph of the introduction section to help readers understand the importance of urban-rural income gap to sustainable development better.

5/ The introduction section adds some introductions on the relationship between the urban-rural income gap and sustainable development. These introductions illustrate that examples from China provide a reference for formulating transportation infrastructure and informatization strategies to promote urban-rural coordination and sustainable development all around the world..

6/ In the second paragraph of the introduction, we have added more detailed research implications to reflect the importance of the research.

Materials and Methods:

7/ The previously submitted manuscript introduces the calculation methods and selection indicators of explained variable urban-rural income gap, the mediating variable innovation investment, and all control variables in detail. The newly submitted manuscript adds explanatory variables intelligence circulation industry detailed to ensure that our analysis can be reproduced by anyone at any time.

Discussion:

8/ Since the topic of our study is to discuss the impact of intelligence circulation industry on the urban-rural income gap, the main innovation lies in the construction of indicators and logical analysis, rather than the innovation of model methods. So the self-criticism of our study is put in the last paragraph of the Conclusion.

9/ Ethical aspects are indeed an integral part of research. We put the discussion of ethical requirements in the last paragraph of the Discussion. The discussion of ethics effectively adds depth of our article.

10/ In the Discussion section, we have removed a lot of repetition of the analysis results. At the same time, a deeper discussion based on these conclusions is added. We integrate the urban-rural income gap with sustainable development more effectively, and we also fully analyze the impact mechanism between the topic.

As a result, the income of urban residents increased accordingly and led to an increase in the urban-rural income gap. However, with the intelligent development of the circulation industry, the cost of land, rent, and other costs in urban areas have increased; therefore, the diffusion effect gradually increased. Innovation investment will be transferred to rural areas with lower cost to accelerate the speed of intelligent development of the circulation industry, thereby improving overall development and income in these areas. These factors have reduced the urban-rural income gap.

Conclusions:

11/ The Conclusion is shortened in four paragraphs. The first paragraph introduces the research background and the possible channels on the intelligence circulation industry affects the urban-rural income gap. The second paragraph briefly summarizes the main findings of our study. The third paragraph makes reliable recommendations for the results. The last paragraph is the self-criticism to our work.

12/ In the last paragraph of the Introduction, we briefly describe the three main innovations of the study. At the same time, these innovations are also highlighted in the second paragraph of the conclusion. The main innovations of our study are:

  • The present study is not limited to the traditional circulation industry and innovatively explores the impact of the intelligent development of China's circulation industry on the urban-rural income gap.
  • At the theoretical mechanism level, this study analyzes the impact of China's circulation industry intelligence on the urban-rural income gap from the agglomeration and diffusion effect, and reasonably explains the heterogeneity of the impact at different stages.
  • This study also adds innovation investment as a mediating effect in the model to further test the impact of the intelligence circulation industry on the urban-rural income gap.

I look forward to working with the reviewers to move our manuscript closer to publication in Sustainability. The manuscript has been rechecked and the necessary changes have been made in accordance with the reviewers’ suggestions. The revised manuscript have been prepared and given below.

Reviewer 3 Report

"Reducing the income gap between urban and rural areas is an important channel for promoting sustainable economic development." Why? This problem is very important. If the article don't answer the question well, the meaning of the topic would be crashed. However, I never see the answer to the question from the rest of the article. I think the subject is not fashion enough and many recent researches have focused on the subject, i don't think it is good enough to publish in the journal. 

Author Response

Thank you very much for your kind comments and suggestions. First of all, the existing papers express the opinion that narrowing the income gap between urban and rural areas could promote sustainable development. I have added the content of this opinion on the first paragraph of the Introduction, and attached references in the study. Second, our study has a reliable theoretical basis. The reduction of the income gap between urban and rural areas could promote the coordinated development of urban areas and rural areas, and balance the speed of urban and rural development, and thus promote sustainable development. Furthermore, our study is innovative. Current papers always focus on the impact of transportation infrastructure on the urban-rural income gap. Our study combines informatization on their opinion, and at the same time selects innovation investment as an intermediary mechanism, and the analysis is highly innovative.

The main innovations of our study are:

  • The present study is not limited to the traditional circulation industry and innovatively explores the impact of the intelligent development of China's circulation industry on the urban-rural income gap.
  • At the theoretical mechanism level, this study analyzes the impact of China's circulation industry intelligence on the urban-rural income gap from the agglomeration and diffusion effect, and reasonably explains the heterogeneity of the impact at different stages.
  • This study also adds innovation investment as a mediating effect in the model to further test the impact of the intelligence circulation industry on the urban-rural income gap.

I look forward to working with the reviewers to move our manuscript closer to publication in Sustainability. The manuscript has been rechecked and the necessary changes have been made in accordance with the reviewers’ suggestions. The revised manuscript have been prepared and given below.

Round 2

Reviewer 2 Report

I have no more comments.

Author Response

Thank you very much for your serious guidance and comments. We look forward to working with you and the reviewers to move our manuscript closer to publication. The manuscript has been rechecked and the necessary changes have been made in accordance with the reviewers’ suggestions. The revised manuscript has been prepared and attached herewith. 

Reviewer 3 Report

I still think the introduction section is not good writing. the first sentence is China's economy is decreasing, and the second one is narrow the gap?  it is rediculas. I recommand some reference which the author can read and study how to write introduction section.

Is Resource Abundance a Curse for Green Economic Growth? Evidence from Developing Countries

Effect of income and energy efficiency on natural capital demand

Export Trade, Embodied Carbon Emissions, and Environmental Pollution: An Empirical Analysis of China’s High and New Tech Industries

Global value chains, technological progress, and environmental pollution: Inequality towards developing countries

Does female labor share reduce embodied carbon in trade?

Review of hidden carbon emissions, trade, and labor income share in China, 2001–2011

Author Response

Based on the article you provided, we have carefully revised the introduction section in our revised manuscript and cited several articles you recommend. Unlike the article you recommend, the introduction section of our manuscript provides introductory statements regarding the topic as well as the review of related literature.

In the first paragraph of the introduction section in our manuscript, we describe the relationship between the urban-rural income gap and sustainable development and also provide basic information regarding the current situation of the urban-rural income gap in China. In the second paragraph, we point out the gaps in the existing literature, stating that no relevant studies on the intelligentization of the circulation industry have been reported so far. At the same time, we have also highlighted the significance of our study in this paragraph. The citations in these two paragraphs make reference to the literature you recommended. Please, see Lines 27–63 in the revised manuscript.

The third paragraph of the introduction section in our manuscript corresponds to the beginning of the literature review section based on the literature you recommended. Specifically, this paragraph summarizes the factors that affect the urban-rural income gap (Revised manuscript, Lines 64–71). Further, the fourth paragraph introduces the definition of the intelligentization of the circulation industry and summarizes its impact on sustainable development (Revised manuscript, Lines 72–87), while in the fifth paragraph, we discuss the impact on the urban-rural income gap from the perspectives of transportation infrastructure as well as information infrastructure, and also point out that studies in which these two aspects were investigated simultaneously are limited (Revised manuscript, Lines 88–104).

Furthermore, the sixth paragraph summarizes the theoretical basis for the mediation effect of circulation industry intelligence development, stating that differences in the level of circulation industry development also bring about differences in foreign investment. This is the mechanism that brings about the urban-rural income gap (Revised manuscript, Lines 105–124).

Finally, the last paragraph of the introduction section in our manuscript is similar to the last paragraph of the introduction section in the article you recommended. Basically, it introduces the theme, novelty, and main content of the manuscript (Revised manuscript, Lines 125–142).

Overall, the framework of the introduction section of our manuscript is somewhat different from that of the article you recommended, given that it covers both introductory information and the literature review. Further, in our revised manuscript, the first, second, and seventh paragraphs provided introductory information, while the third, fourth, fifth, and sixth paragraphs provide the literature review.

Thank you very much for your careful guidance. We look forward to working with you and the reviewers to move our manuscript closer to publication.